# Efficacy and Safety of OnabotulinumtoxinA 400 Units in Patients with Post-Stroke Upper Limb Spasticity: Final Report of a Randomized, Double-Blind, Placebo-Controlled Trial with an Open-Label Extension Phase

**DOI:** 10.3390/toxins12020127

**Published:** 2020-02-18

**Authors:** Masahiro Abo, Takashi Shigematsu, Hiroyoshi Hara, Yasuko Matsuda, Akinori Nimura, Yoshiyuki Yamashita, Kaoru Takahashi

**Affiliations:** 1Department of Rehabilitation Medicine, The Jikei University School of Medicine, Tokyo 105-8461, Japan; 2Department of Rehabilitation Medicine, Seirei Hamamatsu City Rehabilitation Hospital, Shizuoka 433-8511, Japan; t-shige@sis.seirei.or.jp; 3Department of Rehabilitation Medicine, Kikyogahara Hospital, Nagano 399-6461, Japan; hhara448@icloud.com; 4GlaxoSmithKline K.K., Tokyo 107-0052, Japan; akinori.x.nimura@gsk.com (A.N.); kaoru.2.takahashi@gsk.com (K.T.)

**Keywords:** botulinum toxin, stroke, upper limb spasticity, randomized controlled trial

## Abstract

In many countries, 400 units (U) is the maximum dose of onabotulinumtoxinA available to treat upper limb spasticity, but few studies have demonstrated the optimal use of this dose. In the double-blind phase of this randomized, controlled trial, we compared the efficacy and safety of 400 vs. 240 U onabotulinumtoxinA in patients with post-stroke upper limb spasticity. Both groups received 240 U onabotulinumtoxinA injected in the forearm. An additional 160 U onabotulinumtoxinA (400 U group) or placebo (240 U group) was injected in the elbow flexors. Both groups showed similar muscle tone reduction in the wrist, fingers, and thumb; muscle tone reduction in the elbow flexors was greater in the group treated with onabotulinumtoxinA (400 U group) compared to placebo (240 U group). Functional disabilities improved in both groups. No substantial difference was found in safety profiles. In the subsequent open-label phase, all participants received repeat injections of 400 U onabotulinumtoxinA (target muscles and doses per muscle determined by the physician). Similar efficacy and safety outcomes, as with the 400 U group in the double-blind phase, were confirmed. This final report demonstrates that injection of onabotulinumtoxinA 400 U relieves muscle tone in a wide range of areas and improves functional disabilities; generally, it was well-tolerated, and no new safety concerns were identified. The dosing data in the open-label phase will inform optimal use of onabotulinumtoxinA in clinical practice (ClinicalTrials.gov: NCT03261167).

## 1. Introduction

Spasticity has been historically defined as a motor disorder characterized by a velocity-dependent increase in tonic stretch reflexes (muscle tone) [1]. In clinical practice, the term is often used to describe a wide range of disabling symptoms resulting from muscle overactivity, including continuous muscle stiffness, involuntary contractions, and pain and discomfort, which can negatively impact patient quality of life [2].

Botulinum toxin type A (BoNT/A) is included as part of standard treatment modalities for patients with spasticity [3]. Its efficacy and safety for post-stroke upper limb spasticity have been demonstrated in randomized controlled trials [4]. Several formulations of BoNT/A are commercially available. However, these are not interchangeable with one another due to the different clinical characteristics of each [5]. 

OnabotulinumtoxinA (BOTOX, Allergan plc, Irvine, CA, USA) is a BoNT/A formulation that has been approved and used in more than 90 countries for upper limb spasticity, with 400 units (U) stipulated as the maximum dose for this indication in many countries. In Japan, a phase 3 clinical trial was conducted to evaluate the efficacy and safety of 240 U injected in forearm muscles [6], and the results of this trial led to the approval of 240 U as the maximum dose for upper limb spasticity. In clinical practice, however, many patients require injections in elbow flexors (upper arm muscles) and/or shoulder adductors/internal rotators, which often necessitates doses higher than 240 U in order to properly treat these additional muscles. Therefore, we conducted another phase 3 trial to evaluate the efficacy and safety of 400 U as compared to 240 U, in order to obtain approval for the escalation of the maximum dose.

In the double-blind phase of this phase 3, randomized, placebo-controlled trial, we compared the efficacy and safety of onabotulinumtoxinA 400 vs. 240 U in patients with post-stroke upper limb spasticity. In the subsequent open-label phase, up to 3 injections of onabotulinumtoxinA 400 U were given to all patients that met the re-treatment criteria, with the muscles injected (including shoulder adductors/internal rotators and forearm pronators) and doses per muscle determined at the discretion of the physician, to evaluate the efficacy and safety of repeat injections (Figure 1).

## 2. Results

### 2.1. Patient Demographics and Clinical Characteristics

Of 131 patients who were screened, 124 were randomized (61 in the 400 U group; 63 in the 240 U group) and received the first administration of the study drug. A total of 113 patients completed the 48-week study, while 11 patients (5 in the 400 U group; 6 in the 240 U group) withdrew from the trial. The primary reasons for withdrawal were withdrawal of consent (7 patients), adverse events (AEs) (3 patients), and meeting the criteria for discontinuation (1 patient) (Appendix A).

Patient demographics and clinical characteristics have been previously reported and are shown in Table 1. The proportion of male patients was slightly lower in the 400 U group, otherwise no substantial differences were noted between treatment groups. The Modified Ashworth Scale (MAS) score at baseline in the elbow was 3 in 113 patients and 4 in 11 patients (which were coded as 4 and 5, respectively, for tabulation). Of the 4 domains of the Disability Assessment Scale (DAS), “limb position” was selected most frequently as a principal therapeutic target (44 patients, 35%), followed by “dressing” (35 patients, 28%), “hygiene” (28 patients, 23%), and “pain” (17 patients, 14%). These proportions were similar between treatment groups.

At the time of first injection of the study drug, participants were using several rehabilitation modalities including stretching/range of motion (ROM) exercise by 83 patients (67%); muscle strengthening exercise by 41 patients (33%); task-specific training by 29 patients (23%); positioning aids by 16 patients (13%); splinting/orthoses by 14 patients (11%); and taping by 0 patients (0%). No substantial difference was observed between groups regarding the use of these modalities.

### 2.2. Efficacy

During the double-blind phase, both groups received an injection of 240 U onabotulinumtoxinA in the forearm (wrist, finger, and thumb flexors) and an additional injection of onabotulinumtoxinA 160 U (400 U group) or placebo (240 U group) in the elbow flexors at week 0 (treatment cycle 1). At week 6 (primary endpoint), the proportion of patients with ≥ 1-point reduction in the MAS score (responder rate) in the elbow was 68.9% (42/61) in the 400 U group and 50.8% (32/63) in the 240 U group. The difference between the groups was 18.1%, with a 95% confidence interval of 1.1% to 35.0%. The responder rates in the wrist (68.9% for the 400 U group; 81.0% for the 240 U group), fingers (72.1% for the 400 U group; 81.0% for the 240 U group), and thumb (66.7% for the 400 U group; 68.3% for the 240 U group) were comparable between groups at week 6. During the double-blind phase through week 12, the elbow MAS score was consistently lower in the 400 U group compared with the 240 U group (Appendix A), and the change in score (degree of improvement) was consistently greater in the 400 U group vs. the 240 U group. Conversely, score reductions in the wrist, fingers, and thumb were comparable between groups (Appendix A). 

In the open-label phase, in which all patients received 400 U onabotulinumtoxinA (up to 3 treatment cycles separated by ≥12 weeks), responder rates (proportions of patients demonstrating ≥1-point reduction in the MAS score) were similar at week 6 in each treatment cycle (treatment cycles 2, 3, and 4, respectively) at the elbow (77.6%, 73.4%, and 77.8%), wrist (84.8%, 80.0%, and 84.3%), fingers (77.5%, 74.8%, and 81.8%), and thumb (73.6%, 68.5%, and 79.6%) as in the 400 U group during the double-blind phase. Degrees of improvement in MAS scores for these joints were similar to those found in the 400 group after the first injection, with the greatest improvement noted from week 2 through week 6 (Figure 2). The MAS scores for the forearm pronation and shoulder adduction/internal rotation also decreased in the open-label phase (Appendix A).

In the double-blind phase (treatment cycle 1), the DAS principal therapeutic target score decreased from baseline through week 12 in both groups, showing an improvement in functional disabilities due to spasticity. The improvement was comparable between groups, but the mean (standard deviation (SD)) DAS score was numerically lower in the 400 U group (1.4 (0.82) at week 12) compared with the 240 U group (1.6 (0.89) at week 12). In the open-label phase, the DAS score decreased after each injection of 400 U onabotulinumtoxinA (treatment cycles 2, 3, and 4), showing continued improvement in functional disabilities (Table 2).

All DAS domain scores slightly decreased (demonstrating improvement) from baseline through week 12 in both groups during the double-blind phase (Appendix A). The improvement in “limb position” was slightly greater in the 400 U group than in the 240 U group, whereas improvements in the other domains were similar between groups. In the open-label phase, all DAS domain scores showed similar degrees of improvement as shown in the double-blind phase (Appendix A).

Both in the double-blind phase and the open-label phase, the physician-assessed Clinical Global Impression of Change (CGI) showed positive scores (demonstrating improvement) at week 2. This improvement was sustained through week 6 and had attenuated at week 12 of each cycle (Appendix A). The patient-assessed CGI followed a similar course of improvement and attenuation. No substantial difference was noted between groups whether CGI was assessed by the physician or the patient (Appendix A).

### 2.3. Safety

In the first 12 weeks of the double-blind phase, AE rates were similar between groups (51% (31/61) in the 400 U group and 46% (29/63) in the 240 U group). The most frequently reported AEs were nasopharyngitis (11% (7/61) and 17% (11/63) in the 400 U and 240 U groups, respectively) and fall (11% (7/61) and 3% (2/63) in the 400 U and 240 U groups, respectively). Fall and contusion had incidences at least 5% higher in the 400 U group than the 240 U group and arthralgia, muscle spasms, constipation, and subcutaneous hemorrhage occurred only in the 400 U group (Table 3). None of these common AEs were considered by the physician to be related to the study drug.

During the overall study period, the incidence of AEs was 80% (49/61) in the 400 U group and 83% (52/63) in the 240 U group. The most frequently reported AEs overall were fall (25% (15/61) and 17% (11/63) in the 400 U and 240 U groups, respectively) and nasopharyngitis (16% (10/61) and 24% (15/63) in the 400 U and 240 U groups, respectively). Other common AEs reported in ≥ 3% of patients in either group are shown in Appendix A. None of these AEs were considered to be related to the study drug.

Two drug-related AEs as assessed by the physician occurred during the overall study period. Muscular weakness was experienced by one patient (2% (1/61)) in the 400 U group during the double-blind phase, 8 days after the first treatment, and was considered to be local to the site of injection. Mild injection site swelling was experienced by one patient (2% (1/63)) in the 240 U group during the open-label phase, 2 days after the third injection (onabotulinumtoxinA 400 U) and resolved after 28 days.

Across the overall study period, the incidence of nonfatal serious AEs (SAEs) was 11% (7/61) in the 400 U group and 10% (6/63) in the 240 U group. Nonfatal SAEs occurring in 2 or more patients were pneumonia (one in each group), decreased activity (one in each group), and cerebral hemorrhage (2 in the 400 U group). None of the nonfatal SAEs were considered to be related to the study drug. One fatal SAE, pneumonia, occurred during the double-blind phase in the 400 U group, and was assessed as not related to the study drug.

Three AEs leading to study withdrawal occurred in the 400 U group (5% (3/61)), all of which were reported in the double-blind phase and assessed as SAEs not related to the study drug. The AEs included pneumonia (fatal) and cerebral hemorrhage during the double-blind phase, and Alzheimer’s dementia that occurred 243 days after the first injection in the double-blind phase. No AEs leading to study withdrawal occurred in the 240 U group. During the open-label phase, no patient discontinued due to AEs.

### 2.4. Dosing in the Open-label Phase

In the open-label phase, up to 3 injections of 400 U onabotulinumtoxinA were given to all patients that met the re-treatment criteria. The muscles to be injected and doses per muscle were determined at the discretion of the physician. In total, 117 patients received injections of 400 U in 311 treatment sessions across several joints (Appendix A). The mean dose per muscle varied between 28.5 U (flexor pollicis longus) and 74.0 U (biceps brachii). Biceps brachii, brachialis, flexor carpi radialis, and flexor digitorum superficialis were among the most frequently injected muscles (Table 4; more details in Appendix A).

Some agonist muscles work in conjunction with each other to flex or extend a joint, therefore, the target muscles to be injected for the same deformity can vary from patient to patient. Table 5 shows combinations of muscles frequently injected in the open-label phase of this trial. In the treatment of shoulder adduction/internal rotation, for example, most patients received injections in pectoralis major, with or without an additional injection in latissimus dorsi or teres major. More treatment details are shown in Appendix A.

We also performed an exploratory analysis to see whether the severity of spasticity affected the dose injected in each muscle. In most muscles, patients with a MAS score of 3 or 4 (which were coded as 4 and 5, respectively, for tabulation) prior to injection received higher doses of onabotulinumtoxinA compared with those with a lower MAS score, but the differences between the groups were not large (Appendix A).

## 3. Discussion

The responder rates in the elbow at week 6 after the first injection of the study drug (primary endpoint) were 68.9% in the 400 U group and 50.8% in the 240 U group, both of which were extremely close to the values assumed prior to the initiation of the trial (see “Materials and Methods”). Given that the patients in the 240 U group did not receive onabotulinumtoxinA in their elbow flexors, it may seem unexpected that half of them showed a response; however, studies have shown that injecting saline may relieve muscle overactivity [6], and rehabilitation therapy was not prohibited during the trial, which may have contributed to the elbow tone reduction in the 240 U group. Furthermore, injecting 240 U in the forearm muscles might have alleviated the overactivity of elbow flexors, as was suggested in a report using median nerve block [7]. Regarding the selection of elbow flexors for injection, biceps brachii tends to be the first choice among the elbow flexors in the treatment of elbow flexion deformity [8,9], whereas other studies recommend injecting brachialis instead [10,11]. However, based on the results of this trial, which demonstrated a clinically significant improvement in approximately 70% of patients when the majority of those treated in the elbow received treatment in a combination of flexors, injecting multiple elbow flexors may be worth considering.

In joints other than the elbow, the MAS assessment showed comparable improvement in the 400 U and 240 U groups, which was not surprising considering that both groups received the same treatment in the forearm muscles. Similar improvement was also observed for the DAS assessment in both groups, and little between-group differences were observed except for “limb position.” The injection of 240 U BoNT/A in the forearm muscles has been shown to improve the DAS [6]. In our study, the ability of this scale to detect between-group differences may have been limited, because DAS does not assess outcomes directly related to elbow flexion deformities [12]. The same presumption may apply to the results of the CGI, which is a subjective assessment of a broad range of conditions by the physician and the patient.

Regarding the safety assessment, no substantial difference was observed between groups in the incidence of AEs, but the 400 U group demonstrated a slightly more frequent incidence of falls and contusion after the first injection. It is widely known that post-stroke patients have a high risk of falls, and the occurrence of falls per se was not a surprising finding in this trial of elderly patients with hemiplegia and concurrent lower limb spasticity. Injections in the upper limb have been reported to improve walking speed and/or body balance [13,14], and these effects might have affected the incidence of falls due to increased activity, but the results in the open-label phase do not support this hypothesis, as the rate of falls decreased across both treatment groups. In the treatment of upper limb spasticity, attention should be paid to the possibility of change in gross motor functions, but we believe this possibility does not pose a major issue when physicians select treatment modalities.

In this trial, up to 3 injections of 400 U were evaluated during the open-label extension phase, and improvement of muscle tone and functional disabilities were repeatedly demonstrated, with no attenuation of efficacy. Repeat injections of 400 U were generally well-tolerated, and there were no signs of increasing incidence of AEs. Efficacy and safety were demonstrated in an environment similar to real-world clinical practice, where target muscles and doses per muscle were determined at the discretion of the physician. To the best of our knowledge, this is the first study to show how onabotulinumtoxinA 400 U can be distributed in a variety of upper limb muscles. The dosing data in the open-label phase, including doses per muscle and combinations of muscle frequently injected, will provide valuable information when considering how to use 400 U effectively in clinical practice.

This trial has several limitations inherent in a comparative study. In the double-blind phase, injections in shoulder adductors/internal rotators and forearm pronators were prohibited, thus limiting the individualized optimization of botulinum toxin treatment. Furthermore, as the regimen and frequency/intensity of concomitant rehabilitation therapy could not be changed during the double-blind phase, it was difficult to conduct a combination therapy approach best suited for each patient’s post-treatment conditions. In the open-label phase, on the other hand, all patients received 400 U, and there was no control group.

## 4. Conclusions

The results of this randomized, double-blind, placebo-controlled trial demonstrated that injection of onabotulinumtoxinA 400 U relieves muscle tone in a wide range of areas, and improves functional disabilities; generally, it was well-tolerated, and no new safety concerns were identified. The dosing data in the open-label phase of this trial will inform optimal use of onabotulinumtoxinA in clinical practice.

## 5. Materials and Methods

### 5.1. Study Design

This was a phase 3, multicenter, randomized, double-blind, placebo-controlled clinical trial, conducted at 38 medical institutions in Japan from August 2017 through January 2019. The aim of the study was to evaluate the efficacy and safety of onabotulinumtoxinA 400 U as compared with 240 U for upper limb spasticity in post-stroke patients (ClinicalTrials.gov: NCT03261167). 

The study was conducted in compliance with the Declaration of Helsinki, the Ministerial Ordinance on Good Clinical Practice (GCP) for Drugs, and other relevant laws and regulations. The trial protocol was approved by the institutional review board for all 38 medical institutions including the Jikei University School of Medicine (approved by the Jikei University Hospital IRB for Medicinal Products on August 15, 2017), Seirei Hamamatsu City Rehabilitation Hospital (approved by Nakameguro Atlas Clinic IRB on July 5, 2017), and Kikyogahara Hospital (approved by Sugiura Clinic IRB on August 18, 2017). Written informed consent was obtained from all patients.

The study design is shown in Figure 1. The trial consisted of a screening phase (up to 4 weeks), a double-blind phase (minimum of 12 weeks), and an open-label phase (maximum of 36 weeks). In the double-blind phase, patients were randomly allocated in a 1:1 ratio to either 400 U onabotulinumtoxinA or 240 U onabotulinumtoxinA. They received a randomization number at registration based on a computer-generated randomization allocation table. The investigator, study staff at the site, and patient were blinded to the study treatment allocated to each individual patient during the overall study period. The sponsor personnel were unblinded after the database freeze for the interim analysis of the first 24 weeks, and had access to the subject level data. Patients who met the re-treatment criteria after week 12 entered the open-label phase and received 400 U onabotulinumtoxinA up to 3 times, with a minimum interval of 12 weeks between injections. Results in the double-blind phase have been published elsewhere as an interim report (Abo, M., et al., Prog. Med. 2019, 39, 1021–1029, written in Japanese).

### 5.2. Patients and Therapeutic Interventions

Male or female patients (aged 20−80 years) with upper limb spasticity due to stroke were eligible if they had at least a 3-month interval since the most recent stroke, a history of prior treatment with onabotulinumtoxinA 240 U, and a sufficient degree of spasticity to warrant administration of onabotulinumtoxinA 400 U. Their MAS scores had to be at least 3 in the elbow and at least 2 in the wrist or fingers. Exclusion criteria included the presence of bilateral spasticity and fixed contracture in the upper limb to be treated.

At the start of the double-blind phase (week 0), patients in both groups received an injection of 240 U in 6 forearm muscles (flexor carpi radialis, flexor carpi ulnaris, flexor digitorum profundus, flexor digitorum superficialis, flexor pollicis longus, and adductor pollicis). At the same time, an additional injection of onabotulinumtoxinA 160 U (400 U group) or placebo (240 U group) was given, in 3 elbow flexors (biceps brachii, brachialis, and brachioradialis). The recommended doses were 50 U each for flexor carpi radialis, flexor carpi ulnaris, flexor digitorum profundus, and flexor digitorum superficialis; 20 U each for flexor pollicis longus and adductor pollicis; 70 U for biceps brachii; 45 U each for brachialis and brachioradialis, but doses could be adjusted across muscles at the physician’s discretion based on the severity of symptoms, as long as the total dose was 240 U in the forearm muscles and 160 U in the elbow flexors. If the patient had no symptoms in the thumb and 240 U could be divided among the other forearm muscles, the thumb muscles did not need to be treated. The study drug was reconstituted with 2 mL normal saline per 100 U. In order to ensure precise localization of muscles, use of guidance such as electromyography, electrical stimulation, or ultrasonography was recommended.

Concomitant use of centrally acting muscle relaxants was allowed if the patient had been using them before participating in the trial, but the dose was not to be changed from the screening phase until the completion of the double-blind phase. Concomitant use of other botulinum toxin formulations and peripherally acting muscle relaxants (such as dantrolene), nerve blocks with phenol or ethanol, and surgical interventions for the upper limb were prohibited from the screening phase until the completion of the trial. In the double-blind phase, the regimen and frequency/intensity of concurrent rehabilitation therapy were not to be changed.

Entry criteria for the open-label phase included a MAS score of at least 2 in the elbow, and at least 2 in the wrist or fingers, and no drug-related SAE during the double-blind phase. In the open-label phase, patients received 400 U of onabotulinumtoxinA up to 3 times with a minimum interval of 12 weeks between injections. Target muscles were selected from a list of 17 muscles including forearm pronators and shoulder adductors/internal rotators (Table 4). The dose for each of these muscles was determined by the physician, who was provided with a list of recommended doses based on clinical guidelines as reference [15].

### 5.3. Evaluation

The MAS, DAS, and CGI were used for efficacy assessment. Evaluation was performed at weeks 0, 2, 4, 6, and 12. Efficacy analysis included all patients who received the study drug and had at least one assessment of efficacy.

The MAS is a rating scale widely used for spasticity assessment in clinical practice, which measures the degree of resistance to passive joint movement at rest on a scale of 6 levels, from 0 (no increase in muscle tone) to 4 (affected part[s] rigid in flexion or extension) [16]. In this trial, the primary endpoint was the proportion of patients with ≥1-point reduction in the elbow MAS score at week 6 after the first injection. In the tabulation of MAS scores, scores of 0, 1, 1+, 2, 3, and 4 were coded as 0, 1, 2, 3, 4, and 5, respectively.

The DAS is a rating scale to assess the degree of upper limb functional disabilities due to spasticity, which measures 4 domains (“hygiene,” “pain,” “dressing,” and “limb position”) on a scale of 4 levels, from 0 (no disability) to 3 (severe disability) [12]. Before the administration of the study drug, the physician and the patient had a discussion and selected one of the 4 domains as the principal therapeutic target.

The CGI is a global rating scale used by the physician and the patient independently to measure clinical meaningfulness of changes in symptoms on a scale of 9 levels, from −4 (very much worsened) to +4 (very much improved). At the time of assessment by the physician, the patient’s clinical symptoms, AEs, and therapeutic effects to the elbow, wrist, and fingers were considered.

Safety assessment included all patients who received the study drug. AEs were documented throughout the study. Safety endpoints included AEs, vital signs (blood pressure, pulse rate, body temperature), and laboratory tests (hematology, clinical chemistry, urinalysis).

### 5.4. Statistical Analysis

Approximately 120 patients (60 per group) were planned to be enrolled to confirm that the responder rate based on the elbow MAS score in the 400 U group would exceed that in the 240 U group. Assuming that the responder rate in the 400 U group and the 240 U group in this study would be 70.0% and 50.0%, respectively, on the basis of the results of a phase 3 trial conducted in the US [17], the probability of the responder rate in the 400 U group exceeding that in the 240 U group as the point estimate was calculated to be 98% or more if 60 patients were enrolled in each group.

The between-group difference and 95% confidence interval were calculated for the primary endpoint (responder rate in the elbow at week 6 after the first injection). Descriptive statistics were determined for the secondary endpoints (MAS scores and their changes from baseline in the elbow, wrist, fingers, and thumb; DAS principal therapeutic target scores and their changes from baseline). Changes in MAS scores and DAS principal therapeutic target scores were analyzed using the mixed model for repeated measures (MMRM). This model included treatment, visit, and treatment-by-visit interaction as categorical fixed effect, and baseline value and baseline-by-visit interaction as continuous fixed effect. Descriptive statistics were determined for DAS domain scores and the CGI by the physician and the patient. Regarding the exposure to onabotulinumtoxinA in the open-label phase, a post hoc analysis was performed using a negative binomial model with only intercept term to calculate the mean injection rate for each joint and muscle, and a mixed model with only intercept term to calculate descriptive statistics for dosages per joint and muscle. Furthermore, a mixed model with intercept and a MAS score (≥ 3 or <3) prior to the associated dose as a categorical variable was used to evaluate an impact of the prior MAS score on the next dose per muscle. Statistical analyses were performed using SAS version 9.4 (SAS Institute Inc, Cary, NC, USA).

## Figures and Tables

**Figure 1 toxins-12-00127-f001:**
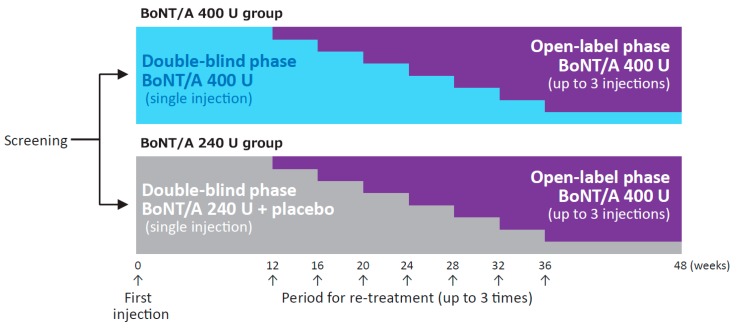
Study design. BoNT/A, botulinum toxin type A (onabotulinumtoxinA). Reproduced with permission from Abo, M., et al., Progress in Medicine, published by Life Science Co., Ltd., 2019 [in Japanese].

**Figure 2 toxins-12-00127-f002:**
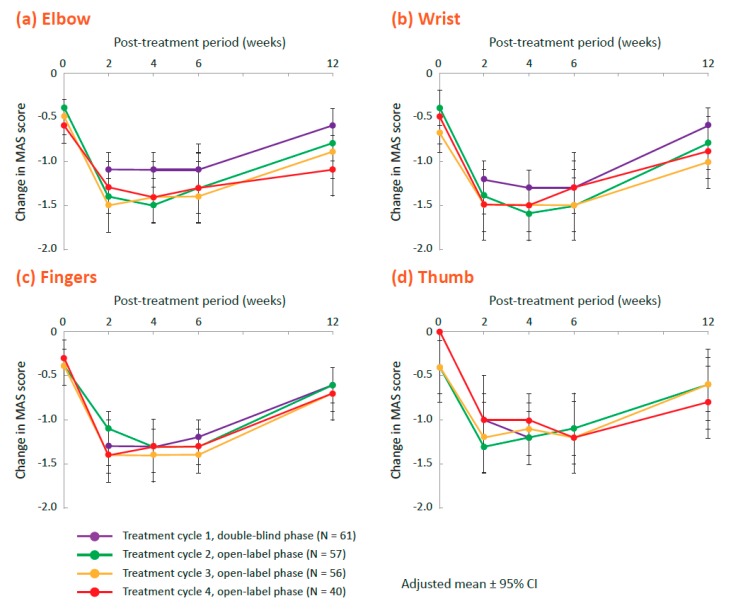
Change from baseline in the Modified Ashworth Scale (MAS) scores across all treatment cycles for the 400 U group at (**a**) elbow, (**b**) wrist, (**c**) fingers, and (**d**) thumb. Treatment cycles are represented as: treatment cycle 1 (double-blind phase, purple), treatment cycle 2 (open-label phase, green), treatment cycle 3 (open-label phase, yellow), and treatment cycle 4 (open-label phase, red). CI, confidence interval.

**Table 1 toxins-12-00127-t001:** Patient demographics and clinical characteristics.

Variables	400 U Group (N = 61)	240 U Group (N = 63)	Total (N = 124)
Sex (male)	46 (75%)	53 (84%)	99 (80%)
Age (years)	57.1 ± 9.90	57.3 ± 10.98	57.2 ± 10.42
Height (cm)	166.2 ± 8.94	164.8 ± 7.31	165.5 ± 8.15
Weight (kg)	67.67 ± 12.794	66.13 ± 10.667	66.89 ± 11.739
Elbow MAS score	4.1 ± 0.28	4.1 ± 0.30	4.1 ± 0.29
DAS principal therapeutic target score	2.0 ± 0.80	1.9 ± 0.87	2.0 ± 0.84

Number of patients (%) or mean ± standard deviation. DAS, Disability Assessment Scale; MAS, Modified Ashworth Scale. MAS scores of 0, 1, 1+, 2, 3, and 4 were coded as 0, 1, 2, 3, 4, and 5, respectively, for tabulation: 0 = no increase in muscle tone; 5 = affected part(s) rigid in flexion or extension. Reproduced with permission from Abo, M., et al., Progress in Medicine, published by Life Science Co., Ltd., 2019 [in Japanese].

**Table 2 toxins-12-00127-t002:** Change from baseline in the DAS principal therapeutic target score across all treatment cycles for the 400 U group.

Treatment Cycles	Parameters	Week 0	Week 2	Week 4	Week 6	Week 12
**Double-blind phase**						
First injection	Mean	−	−0.6	−0.6	−0.7	−0.6
(N = 61)	95% CI	−	(−0.8, −0.4)	(−0.8, −0.5)	(−0.9, −0.5)	(−0.9, −0.4)
**Open-label phase**						
Second injection	Mean	−0.6	−0.7	−0.8	−0.8	−0.7
(N = 57)	95% CI	(−0.8, −0.4)	(−0.9, −0.5)	(−1.0, −0.6)	(−1.0, −0.6)	(−0.9, −0.5)
Third injection	Mean	−0.6	−0.9	−0.9	−0.8	−0.8
(N = 56)	95% CI	(−0.8, −0.4)	(−1.1, −0.6)	(−1.1, −0.6)	(−1.1, −0.6)	(−1.0, −0.5)
Fourth injection	Mean	−0.7	−0.9	−0.9	−0.9	−0.9
(N = 40)	95% CI	(−1.0, −0.4)	(−1.2, −0.6)	(−1.2, −0.6)	(−1.1, −0.6)	(−1.1, −0.6)

CI, confidence interval; DAS, Disability Assessment Scale.

**Table 3 toxins-12-00127-t003:** Common adverse events in the 12 weeks following the first injection (double-blind phase).

Adverse Events	400 U Group (N = 61)	240 U Group (N = 63)
Total number of patients with adverse events	31 (51%)	29 (46%)
Nasopharyngitis	7 (11%)	11 (17%)
Fall	7 (11%)	2 (3%)
Contusion	4 (7%)	1 (2%)
Influenza	2 (3%)	3 (5%)
Insomnia	3 (5%)	1 (2%)
Oxygen saturation decreased	3 (5%)	1 (2%)
Back pain	1 (2%)	2 (3%)
Arthralgia	2 (3%)	0
Muscle spasms	2 (3%)	0
Constipation	2 (3%)	0
Hemorrhage subcutaneous	2 (3%)	0

Number of patients (%). Adverse events reported by more than one patient in either group are shown. Reproduced with permission from Abo, M., et al., Progress in Medicine, published by Life Science Co., Ltd., 2019 [in Japanese].

**Table 4 toxins-12-00127-t004:** Muscles injected in the open-label phase.

Muscles	No. of Patients	No. of Treatment Sessions	Mean Dose (95% CI)
All muscles combined	117	311	−
Pectoralis major	84	195	56.2 (51.5, 60.8)
Latissimus dorsi	36	69	45.8 (38.4, 53.1)
Teres major	16	27	42.1 (34.9, 49.3)
Subscapularis	9	17	42.7 (33.9, 51.4)
Biceps brachii	116	296	74.0 (70.4, 77.6)
Brachialis	110	261	45.3 (42.6, 47.9)
Brachioradialis	87	168	41.6 (38.9, 44.2)
Pronator teres	73	175	47.2 (43.9, 50.5)
Pronator quadratus	4	8	45.3 (15.4, 75.2)
Flexor carpi radialis	108	267	50.1 (46.9, 53.2)
Flexor carpi ulnaris	94	213	42.4 (40.1, 44.6)
Flexor digitorum profundus	85	200	48.8 (45.0, 52.6)
Flexor digitorum superficialis	111	276	56.2 (52.9, 59.5)
Lumbricals	22	46	39.6 (31.1, 48.1)
Flexor pollicis longus	84	174	28.5 (26.1, 30.9)
Adductor pollicis	80	166	28.9 (26.0, 31.8)
Opponens pollicis	7	12	−

The estimate for the mean dose in opponens pollicis was not obtained because of the small amount of data. CI, confidence interval.

**Table 5 toxins-12-00127-t005:** Combinations of muscles frequently injected in the open-label phase.

Joints	Muscles	No. of Patients	No. of Treatment Sessions	Mean Dose (95% CI)
Shoulder	PM	58	111	55.3 (50.7, 60.0)
	PM + LD	30	54	101.0 (84.8, 117.1)
	PM + TM	6	10	−
Elbow	BB + B + BR	74	147	160.2 (151.8, 168.6)
	BB + B	58	106	117.7 (109.7, 125.6)
	BB	25	28	89.6 (81.2, 98.1)
	BB + BR	12	15	125.0 (101.7, 148.4)
Forearm (pronation)	PT	71	167	47.1 (43.8, 50.5)
Wrist	FCR + FCU	93	210	91.2 (86.0, 96.5)
	FCR	36	57	53.7 (50.2, 57.2)
Fingers	FDP + FDS	75	177	102.9 (95.9, 109.9)
	FDS	37	66	60.7 (53.8, 67.5)
	FDP + FDS + LI	10	18	136.6 (118.4, 154.9)
	FDS + LI	9	15	110.3 (92.5, 128.1)
	LI	6	10	−
Thumb	FPL + AP	55	111	52.2 (46.9, 57.6)
	FPL	38	56	30.6 (26.9, 34.4)
	AP	33	48	37.8 (32.5, 43.0)

AP, adductor pollicis; B, brachialis; BB, biceps brachii; BR, brachioradialis; CI, confidence interval; FCR, flexor carpi radialis; FCU, flexor carpi ulnaris; FDP, flexor digitorum profundus; FDS, flexor digitorum superficialis; FPL, flexor pollicis longus; LD, latissimus dorsi; LI, lumbricals; PM, pectoralis major; PT, pronator teres; TM, teres major. Some estimates were not obtained because of the small amount of data.

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
