# Peer review of "Efficacy and Safety of OnabotulinumtoxinA 400 Units in Patients with Post-Stroke Upper Limb Spasticity: Final Report of a Randomized, Double-Blind, Placebo-Controlled Trial with an Open-Label Extension Phase"

_toxins, 2020, doi:10.3390/toxins12020127_

Round 1

Reviewer 1 Report

I read the paper entitled “Efficacy and safety of Onabotulinumtoxin A 400 Units in Patients with post-stroke Upper limb spasticity: a randomized, double-blind, placebo-controlled trial with open-label extension phase” submitted to this Journal for publication.

The manuscript describes a study aimed to evaluate the efficacy and safety of the treatment of post-stroke upper limb spasticity with 400 U onabotulinumtoxinA as compared to 240 U.

This evaluation was carried on 124 screened patients randomized in two groups, one receiving 400 U (240U in 6 forearm muscles + 160 in elbow flexors) of  onabotulinumtoxinA, the other 240 U (forearm muscles) plus placebo (elbow flexor muscles). The double-blind phase was followed by a open-label phase with the administration of 400U of onabotulinumtoxinA up to three times (list of 17 muscles).

The results obtained form double-blind trial show that elbow flexors tone reduction was significantly grater  400U group with respect to 240U group, although no differences in muscle tone reduction was observed in the other treated muscles. Functional disabilities were improved in both groups and safety profiles were similar in both groups, although a higher tendency to falls and contusions were observed in 400U group. Functional improvements and safety profiles observed in double-blind phase were confirmed in open-label extension phase using repeated injections of 400U.

The study appear to be robust and it is a phase 3 completed clinical trial (ClinicalTrials.gov #03261167).

Results are well described. Materials and methods are clear in almost all sections. I would give a more in deep description of utilized statistical methods (indeed are very well described in the ClinicalTrial.gov site) especially as regard the Mmodels and the utilized software to analyse them.

Discussion requires in my opinion some comments:

I have some concerns about the possibility that a saline injection relieves muscle hypertone. Could be possible, through a local re-equilibrium of extracellular ion concentration (potassium for its influence on membrane potential, calcium for its still discussed effect on membrane biophysical excitability). However it is hard to think that this single administration could be effective for a long period of time, if the causal factor is still present. Nevertheless it is possible the opposite, i.e. that a causal factor evokes a physiopathological loop which auto-maintains itself even when the primary causal factor is lost. In this second case, however, I find difficult not to consider to use saline as a therapeutical agent instead of 400U onabotulinumtoxinA. Obviously, considering the robustness of the study, the above mentioned hypotheses (and maybe others) should be better discussed finding, I think, a more convincing explanation for these particular findings. I understand the explanation regarding falls and contusions. The question is: should they be considered in evaluating pros and cons ? I would authors to deeply going in evaluating this aspect.

In conclusion I think that the MS can be considered for publication in Toxin after a moderate revision on the Statistical Analysis section of Material and Methods and on Discussion.

Minor Comments:

Some tables appear to be splitted in two pages

Figure 2 colors are confusing with respect to figure 1. I would have utilized gray color for 240U and blue for 400U group

Reference 6, repeatedly cited, is absent or not accessible.

Author Response

Thank you very much for your comments. We added some descriptions on statistical models and the software used (in "Materials and Methods"). Regarding tone reduction in the 240 U group, we suspect that several factors were involved such as saline injections in elbow flexors, botulinum toxin injections in the forearm, and concomitant therapy (newly added). We would like to just refer to them as possible causes, instead of discussing which one had the greatest impact, since this is not a new finding and there is no way to confirm the hypotheses. In "Discussion," we also added our comment on falls.

Response to your minor comments:

We believe that the editor of Toxins will take care of the appearance of the tables; Figure 2 has been deleted in response to comments by another reviewer; The former "Reference 6" (now deleted) is an interim report of this clinical trial written in Japanese, and published in a local Japanese journal. Since the editor of Toxins has told us that we can only cite articles in English, it is now gone and details of the interim report are described in the legends of Figure 1 and Tables 1 & 3, and the text ("Materials and Methods").

Reviewer 2 Report

The purpose of this study is not clearly described whether dose of '400 U' is an issue or additional injection of elbow flexor muscle is an issue. 

What is the reason to determine the dose of 240 U at the forearm? 

What is the reason to determine additional dose of 160U at the forearm?

Why authors did not describe detailed information of the dose and the muscle of double blind study where as they described the detailed information in Table 4?

What is the meaning of 'No. of treatment session' in Table 4.

There  is no information of ref. 6. 

Author Response

Thank you very much for your comments. We added some background information of this clinical trial (in "Introduction") to show how it came to be conducted, why 240 U was injected in the forearm, and why 160 U was injected in the elbow flexors. We also added detailed information on dosing in the double-blind phase (in "Materials and Methods").

"No. of treatment sessions" in Table 4 shows how many times each of these muscles was treated in the open-label phase. The text above the table says "117 patients received injections of 400 U in 311 treatment sessions," and we put these figures in the table to facilitate understanding of readers.

The former "Reference 6" (now deleted) is an interim report of this clinical trial written in Japanese, and published in a local Japanese journal. Since the editor of Toxins has told us that we can only cite articles in English, it is now gone and details of the interim report are described in the legends of Figure 1, Table 1 and Table 3, and the text ("Materials and Methods").

Reviewer 3 Report

My biggest concern with this paper is the data analysis.  I do not understand why the authors chose the analyses they did, but they are no appropriate.  There are no statistical analyses that determine whether or not the differences between groups are statistically significant.  Without this, the information is useless.  The analysis is not complete.  The authors commented on the size of difference between groups, but this is meaningless without determining whether or not it is statistically significant.  The analysis needs to be redone and the manuscript rewritten.

Author Response

Thank you very much for your comments.

As described in this revised version of the manuscript, this clinical trial was conducted to obtain approval for the escalation of the maximum dose for upper limb spasticity in Japan. The study protocol, therefore, was created through discussions with the authoritative agency, and it was designed to confirm that a point estimate of the responder rate would be higher in the 400 U group as compared to the 240 U group (which means that the sample size was not based on power).

The study protocol (including a primary analysis plan) was fixed before the initiation of the trial. Performing an unplanned statistical testing after the trial does not seem to be an appropriate way to scientifically evaluate the efficacy results. Although no statistical testing was performed, efficacy can be evaluated objectively with point estimates and confidence intervals, and all the procedures are clearly described in the manuscript. We believe this paper deserves publishing in a scientific journal, and the results demonstrated here would be of great interest to readers of Toxins.

Reviewer 4 Report

Manuscript for „Toxins“ versus „Pro. Med” (already published in Japanese):

Title: very similar.
Abstract: very similar.

Study overview: the same kind of study (randomized, double-blind, placebo-controlled trial); the same phase of the study (12 weeks); the study conducted at same 38 medical institutions; the same study duration; the same clinical trials permission number.
The same description of the trials.
The same patient’s disposition.
Absolutely the same doses of BoNT-A used in both studies: BoNT-A 400 U vs 24o U BoNT-A .
The same main conclusion of both papers: no substantial differences was noted for the safety profile of BoNT-A 400 U as compared with that of BoNT-A 240 U.

Patients and therapeutic interventions: absolutely the same patients; the same patient’s history, the same description.
The same demographic and clinical characteristics of the patients.

Evaluation: the same scales of the evaluation and the same description.

Statistical analysis: absolutely the same.
The same injected parts of the body.
The same injected muscles.

Results:
Patient’s background: the same patients, the same rating scale.
Efficiency: the same efficiency.
Safety: the same safety.

Figure 1.: (Study design): only very little difference.
Figure 2.: Absolutely the same.
Table 3.: Absolutely the same.

Conclusion: both publications are very similar. Shortly to say: manuscript for publishing in “Toxins” is nearly the same with the previously publication of the same group in „Pro. Med” (in Japanese).
Comparing the both manuscript that I have under review and the translated journal article, it is clear that the toxins-690834 is a more or less modified (very similar) version of the former article already published in Japanese.

The manuscript send to Toxins has no new or interestimg results not already shown in the published article of „Pro. Med”.

Author Response

Thank you very much for your comments.

The former "Reference 6" (now deleted) is an interim report of this clinical trial, and focuses on findings in the first 12 weeks (double-blind phase) of the study. It was written in Japanese, and published in a local Japanese journal. Since the editor of Toxins has told us that we can only cite articles in English, it is now gone and details of the interim report are described in the text ("Materials and Methods") and the legends of figures/tables.

The manuscript being reviewed, on the other hand, is a final report of the 48-week trial; it contains findings during the entire period, including information on the safety and efficacy of repeat injections, as well as results of an additional analysis on dosing in the open-label phase.

In order to clearly show differences described above, we changed the title of this manuscript to indicate that this is a final report, and added an explanation about the interim report (in "Materials and Methods"). We also deleted one of the figures that had appeared in the interim report. The revised manuscript features Figure 2, Table 2, and Tables S1-S3 to demonstrate efficacy results of repeated injections, and Table S4 (which seems to be too large to include in the main part) shows safety results during the entire period, all of which have not been published before. Along with the results of an additional analysis, we believe this final report will be of great interest to readers of Toxins, especially those outside Japan with no access to the interim report.

Round 2

Reviewer 2 Report

1. Title is not appropriate

Authors studied the efficacy and safety of additive 160 unit injection to proximal arm muscle comparing with 240 unit injection to forearm muscle in post stroke population.

Current title does not represent the main purpose and outcome of this study. 

2.  Detailed result comparing 400 unit and 240 unit should be the main result of efficacy. But authors does not show detailed statistical result of MAS and DAS comparing two groups neither in table 2 nor S1.

3. Without the the detailed superior efficacy of 400 units, conclusion can be made. Rather no additional efficacy 400 unts may be the coclusion.  

Author Response

Thank you very much for your comments.

In this trial, efficacy was evaluated not only in elbow flexors (which received 160 units), but also in wrist and finger flexors (which received 240 units). Moreover, using as much as 400 units in a single limb could lead to adverse events not associated with lower doses, and safety assessment was one of the main purposes of this trial. Therefore, we believe the title should state "efficacy and safety of 400 units," instead of 160 units.

As described in Introduction, this trial was conducted to obtain approval for the escalation of the maximum dose for upper limb spasticity in Japan, and the study protocol was created through discussions with the regulatory agency. The protocol (including a statistical analysis plan) was fixed before the initiation of the trial, and it was designed to confirm that a point estimate of the responder rate would be higher in the 400 U group as compared to the 240 U group. Although no statistical testing was performed, efficacy can be objectively evaluated with point estimates and confidence intervals, and the confidence interval of the primary endpoint (1.1-35.0%) was higher than 0%, showing superiority of 400 units over 240 units. With all these statistical procedures clearly described in the manuscript, we believe this paper deserves publishing in a scientific journal, and the results demonstrated here would be of great interest to readers of Toxins.

Reviewer 4 Report

This second version of the ms has a bit different title; in the abstract, however, not a single word was changed.

Now it is clear that more or less of the present ms has already been published in a local journal.

However, compared with the translated Version (in Japanese) the authors added further data (especially Tables), so that the changed title ... Final report .. gives some sense.

Author Response

Thank you very much for your comments. We also edited the abstract to show that this is a final report.